# Child Anthropometrics and Neurodevelopment at 2 and 3 Years of Age Following an Antenatal Lifestyle Intervention in Routine Care—A Secondary Analysis from the Cluster-Randomised GeliS Trial

**DOI:** 10.3390/jcm11061688

**Published:** 2022-03-18

**Authors:** Monika Spies, Kristina Geyer, Roxana Raab, Stephanie Brandt, Dorothy Meyer, Julia Günther, Julia Hoffmann, Hans Hauner

**Affiliations:** 1Institute of Nutritional Medicine, Else Kröner-Fresenius-Centre for Nutritional Medicine, School of Medicine, Technical University of Munich, Georg-Brauchle-Ring 62, 80992 Munich, Germany; 2Centre for Hormonal Disorders in Children and Adolescents, Division of Paediatric Endocrinology and Diabetes, Department of Paediatrics and Adolescent Medicine, Ulm University Medical Centre, Eythstraße 24, 89075 Ulm, Germany; 3European Foundation for the Care of Newborn Infants (EFCNI), Hofmannstraße 7a, 81379 Munich, Germany

**Keywords:** child development, anthropometrics, neurodevelopment, antenatal lifestyle intervention, routine care, obesity prevention

## Abstract

Maternal characteristics around pregnancy may influence obesity risk and neurodevelopment in children. To date, the effect of antenatal lifestyle interventions on long-term child development is unclear. The objective was to investigate the potential long-term effects of an antenatal lifestyle intervention programme conducted alongside routine care on child anthropometrics and neurodevelopment up to 3 years of age. Mother-child pairs from the cluster-randomised GeliS trial were followed up to 3 years of age. Data on child anthropometrics in both groups were collected from routine health examinations. Neurodevelopment was assessed via questionnaire. Of the 2286 study participants, 1644 mother-child pairs were included in the analysis. Children from the intervention group were less likely to score below the cut-off in Fine motor (*p* = 0.002), and more likely to have a score below the cut-off in Problem-solving (*p* < 0.001) compared to the control group at 3 years of age. Mean weight, height, head circumference, body mass index, and the respective z-scores and percentiles were comparable between the groups at 2 and 3 years of age. We found no evidence that the lifestyle intervention affected offspring development up to 3 years of age. Further innovative intervention approaches are required to improve child health in the long-term.

## 1. Introduction

According to the World Health Organization, 39 million children worldwide under the age of 5 were affected by overweight or obesity in 2020 [1]. Similarly, there is a high prevalence of childhood overweight and obesity in Germany [2]. These numbers are alarming, since it is known that children with obesity are five times more likely to have obesity in adulthood [3], and have an increased risk of morbidities, such as type 2 diabetes, coronary heart disease, and cancer, later in life [4]. Furthermore, neurodevelopmental disorders seem to be more prevalent in children with obesity [5].

Several maternal factors are thought to influence child anthropometrics and neurodevelopment. For example, excessive gestational weight gain (GWG) has been discussed to be associated with an increased risk for childhood overweight and obesity [6]. A healthy maternal lifestyle during pregnancy [7], as well as breastfeeding [8], seem to decrease the child’s obesity risk. Maternal nutrition, especially in late pregnancy [9], has been shown to impact child neurodevelopment, whereas breastfeeding duration seems to be linked to better cognitive and motor development [10]. Furthermore, maternal pre-pregnancy overweight and obesity appear to be associated with both an increased risk of offspring overweight and obesity [11,12], as well as neurodevelopmental problems [13,14,15]. In their meta-analysis, Sanchez et al. found the risk for adverse neurodevelopmental outcomes to be 17% higher in children of mothers with overweight, and 51% higher in children of mothers with obesity before pregnancy [15].

Multiple studies have aimed at improving maternal lifestyle during pregnancy, and limiting GWG through diet- and/or physical-activity-based interventions [16]. The moderate effectiveness of lifestyle interventions in reducing GWG has been shown in an individual participant data meta-analysis [16], as well as in a recent meta-review [17].

However, the evidence for the long-term influence of maternal lifestyle interventions on offspring overweight and obesity risk is less clear [18], and the results of studies investigating their impact on child neurodevelopment are heterogeneous [19,20,21,22]. Two recent meta-analyses found that lifestyle interventions in pregnancy had no impact on either short- or long-term weight and growth outcomes in children [23], nor on early childhood obesity risk [24]. However, most studies had rather small sample sizes and were performed in community settings or academic institutions. So far, the effect of a large-scale lifestyle intervention during pregnancy conducted in a real-life setting, and including women with normal weight, overweight, and obesity, on child long-term development has not been investigated.

The “Gesund leben in der Schwangerschaft” (“healthy living in pregnancy”) (GeliS) trial, which aimed at supporting long-term maternal and child health through an improved maternal health behaviour, intended to fill these knowledge gaps. This large-scale cluster-randomised trial implemented a lifestyle intervention embedded in routine antenatal care in women with normal weight, overweight, and obesity. The trial was not successful in influencing the primary outcome GWG [25], but yielded improvements in maternal lifestyle [26,27] extending into the first year postpartum [28]. No major modification of infant growth was detected in the first year of life [29]. Though patterns of complementary feeding were mainly comparable between groups [29], there was a slightly higher rate of exclusive breastfeeding in the intervention group (IG) [30]. In this secondary analysis, we aimed at investigating the potential long-term effects of the GeliS lifestyle intervention on child anthropometrics and neurodevelopment in the 2nd and 3rd years of life.

## 2. Materials and Methods

### 2.1. The GeliS Study: Design and Setting

The GeliS study is a prospective, cluster-randomised, controlled, open intervention trial conducted in five regions in Bavaria, Germany. Two districts per region were selected and randomised to intervention and control arms. Within the intervention and control districts, study participants were recruited in gynaecological and midwifery practices. In the intervention districts, lifestyle counselling was conducted in the practices alongside the German routine antenatal care visits. Details on the study design and cluster-randomisation have been published previously [31]. The primary aim of the study was to reduce the proportion of women with excessive GWG, defined according to the Institute of Medicine (IOM) criteria [32]. Details on primary, as well as secondary, outcomes have already been published [25,26,27,28,29,30,33,34].

The study complied with local regulatory requirements, and was conducted in conformity with the Declaration of Helsinki. The Ethics Commission of the Technical University of Munich endorsed the study protocol (project number 5653/13), and the study was registered in the ClinicalTrials.gov Protocol Registration System (accessed on 18 November 2021) (NCT01958307).

### 2.2. Participants

In total, 71 participating gynaecological and midwifery practices recruited eligible pregnant women between 2013 and 2015. The participants had to meet the following inclusion criteria: pre-pregnancy body mass index (BMI) ≥18.5 kg/m^2^ and ≤40.0 kg/m^2^, age between 18 and 43 years, <12th week of gestation, singleton pregnancy, sufficient command of the German language, and provision of informed consent. Women were excluded from the study if they suffered severe complications which compromised the intervention [31]. Throughout the follow-up phase, women were regarded as drop-outs if they did not provide contact details, could no longer be reached, or withdrew participation [30].

### 2.3. Lifestyle Intervention

Women in the IG received a lifestyle intervention programme, which consisted of four face-to-face counselling sessions in the participating practices, carried out alongside routine care visits by previously trained medical personnel, midwives, or gynaecologists. Three counselling sessions were provided during pregnancy (12th–16th, 16th–20th, and 30th–34th week of gestation), and one after birth (6–8 weeks postpartum). The counselling content comprised information on an adequate GWG according to the IOM recommendations [32], recommendations on healthy dietary and physical activity behaviour according to national and international recommendation ns [35,36], the value of a balanced ante- and postnatal lifestyle and breastfeeding for healthy offspring development, and supportive information for introducing complementary feeding. Details on the lifestyle intervention programme and counselling content have been published in the study protocol [31]. Women in the control group (CG) received routine care with the addition of a flyer containing information on a healthy lifestyle during pregnancy, and breastfeeding recommendations.

Women from both the CG and IG were included in a 5-year follow-up observation programme, during which child, as well as maternal, data were collected 1, 3, and 5 years after birth via phone interviews and questionnaires.

### 2.4. Data Collection and Processing

Maternal sociodemographic data were collected via a screening questionnaire at recruitment (<12th week of gestation). Pre-pregnancy BMI was calculated using self-reported pre-pregnancy weight and height. Maternal weight during pregnancy was retrieved from maternity records. The last measured weight before delivery, and the first measured weight at recruitment, were used to calculate GWG. Infant anthropometrics at birth, as well as the exact birth date, stemmed from birth records. Data on current maternal smoking status were derived from a questionnaire filled out by the participants 3 years after birth.

Data on infant anthropometrics in the 1st year of life have already been analysed and published [29]. Child anthropometric data at 2 and 3 years of age were collected from the well-baby check-up booklet used by physicians for the routinely conducted health check-up programme for children in Germany. The two health examinations, scheduled during the 2nd and 3rd years of life, should be performed between 21–24 months old and between 34–36 months old. Information on documented data, including child weight, height, and head circumference, from the two examinations, as well as the dates of the examinations, were collected from the participants via phone interviews conducted around the 3rd birthday of the child. Data from a German reference group were used to calculate sex-specific percentiles and z-scores for age from these measurements [37]. In accordance with German recommendations, a BMI-for-age-percentile below 10.0 was used to group children as having underweight, a percentile above 90.0 as having overweight, and a percentile above 97.0 as having obesity [37]. The child’s exact age at the two health examinations in the 2nd and 3rd years of life was calculated by subtracting the date of birth from the date of the respective examination. In case of missing examination dates, single imputation was applied.

Data on child neurodevelopment were collected at the age of 3 years using the German 36 months version of the Ages and Stages Questionnaire (ASQ-3^TM^) (ASQ). As the German version of the ASQ was not yet available for use in our study at the start of the 3-year follow-up data collection, it could not be handed out to the first 319 participants contacted. The questionnaire was sent to the participants by post, and was completed without study team supervision. The exact age of the child at the time of completion was calculated by subtracting the child’s date of birth from the questionnaire completion date, using single imputation in case of missing questionnaire completion dates. The ASQ comprises five developmental domains (Communication, Gross motor, Fine motor, Problem-solving, and Personal-social). The 6 questions per domain focus on age-appropriate development, and can be answered either “yes” (10 points) if the child has mastered the task in the question, “sometimes” (5 points) if the task is not mastered frequently, or “not yet” (0 points) if the child has not yet mastered the task. According to the user’s guide [38], and as done by others [10,21], the points achieved in each of the five domains were summed up for each child, with a higher score indicating a closer-to-age-appropriate development in that domain. The mean value of the non-missing questions for a domain was inserted in the case of ≤2 missing questions in a domain. If >2 questions were missing, no score was calculated for the domain, and the ASQ was excluded from the analysis of this domain. Additionally, the scores per domain were evaluated for each child using the cut-off values provided by the questionnaire. If children scored lower than a cut-off value of 30.99 points for Communication, 36.99 points for Gross motor, 18.07 points for Fine motor, 30.29 for Problem-solving, or 35.33 points for Personal-social, this indicated a potential delay in development in that area [38].

### 2.5. Statistical Analysis

The power calculation for the GeliS study was conducted for the primary endpoint excessive GWG, as described elsewhere [31]. All participants who provided data on child anthropometrics and/or neurodevelopment were included in the current analyses. Group differences in child anthropometric outcomes were calculated using likelihood-based mixed models for repeated measures, as described by Bell et al. [39]. Analyses included data from all health examinations from the 1st year of life (already published [29]) to the 3rd year of life, and customised hypotheses were applied to investigate group differences at 2 and 3 years of age. By including visit number and group assignment, as well as their interaction, point estimates and 95% confidence intervals were obtained for the mean differences between IG and CG at 2 and 3 years of age. The likelihood-based mixed models for repeated measures were adjusted for maternal pre-pregnancy age, maternal pre-pregnancy BMI category, parity, child sex, child age in days at the corresponding visit, and study region. Adjustment for child age at each visit was used to account for individual deviations from the proposed age for the check-up visits. Analyses of age- and sex-specific percentiles and z-scores were not adjusted for child sex and child age.

Between-group differences in weight categories based on BMI-for-age percentiles at 2 and 3 years of age were assessed using proportional odds ordinal logistic regression models fit with generalised estimating equations (GEEs), as described by Donner at al. [40], and were adjusted for maternal pre-pregnancy age, maternal pre-pregnancy BMI category, and parity.

Differences between the IG and CG in ASQ scores and ASQ score evaluation were analysed using linear and binary logistic regression models fit with GEEs. The models were adjusted for maternal pre-pregnancy age, maternal pre-pregnancy BMI category, maternal educational level, parity, child sex, and child age in months at completion of the ASQ. Again, adjustment for child age at completion was applied to account for deviations from the target age range for the 36 months version of the ASQ.

All analyses were conducted using SPSS software (IBM SPSS Statistics for Windows, version 26.0, IBM Corp, Armonk, NY, USA), and *p*-values < 0.05 were considered statistically significant. Due to the exploratory character of the analyses, no adjustment for multiple testing was performed.

## 3. Results

### 3.1. Participant Flow and Baseline Characteristics

Of the originally allocated 2261 participants (IG: n = 1139, CG: n = 1122), 1783 entered the 3-year follow-up phase (Figure 1). Of these, 63 mother-child pairs in the IG, and 73 mother-child pairs in the CG were lost to follow-up, which amounts to a total drop-out rate of 7.6% for the 3-year follow-up phase. Three participants were excluded due to missing data, leaving 1644 mother-child pairs who were included in this analysis. Data on child anthropometrics and neurodevelopment were provided by 1625 and 1164 participants, respectively.

Table 1 depicts the characteristics of the mother-child pairs included in the analyses, which were largely similar in both groups. Maternal pre-pregnancy age, pre-pregnancy weight, pre-pregnancy BMI, as well as GWG were comparable between the IG and CG. In total, more women had pre-pregnancy normal weight (65.5%), than overweight (22.9%) or obesity (11.6%). As reported previously [25], the rate of primiparous women was higher in the IG than in the CG (63.6% vs. 54.4%). Furthermore, the proportion of male infants was lower in the IG (50.9%) compared to the CG (54.4%). Among the women from the intervention group included in this analysis, 96.4% attended all four counselling sessions (data not shown).

### 3.2. Child Anthropometrics

Table 2 depicts child anthropometric measurements at 2 and 3 years of age. The mean weights were comparable between IG and CG at both time-points (2 years of age: IG 12.30 ± 1.44 kg vs. CG 12.26 ± 1.37 kg, *p* = 0.176; 3 years of age: IG 14.58 ± 1.73 kg vs. CG 14.54 ± 1.72 kg, *p* = 0.166). Mean height, mean head circumference, mean BMI, and the respective mean z-scores and percentiles were similar between IG and CG. The mean BMI percentile at 3 years of age was almost identical in both groups (*p* = 1.000). Furthermore, the odds for a higher weight category at 2 years of age were not significantly different for children from the IG versus CG (*p* = 0.274) (Table 2). However, children of the IG were, by trend, more likely to be in a higher weight category compared to children from the CG at 3 years of age (adjusted OR: 1.17, 95% CI 0.99 to 1.37; *p* = 0.062). Overall, there were no significant differences in child anthropometric measurements between groups at 2 and 3 years of age. In an exploratory sensitivity analysis, the exclusion of preterm births did not affect the results (data not shown). No between-group differences were detected in the subgroup of children from women with pre-pregnancy overweight or obesity (data not shown). Irrespective of maternal pre-pregnancy BMI category, the prevalence of overweight among all children at 3 years of age was 6.6% (IG: 7.0%, CG: 6.2%), and the prevalence of obesity was 1.8% (IG: 1.8%, CG: 1.7%) (Table 2).

### 3.3. Child Neurodevelopment

The child ASQ scores in the five domains at 3 years of age are summarized in Table 3. The scores in Communication (*p* = 0.488), Gross motor (*p* = 0.217), Fine motor (*p* = 0.335), and Personal-social (*p* = 0.744) were comparable in the IG and CG. The IG had significantly lower scores in Problem-solving, but the difference was very small (IG: 54.3 ± 8.1, CG: 54.9 ± 7.2; *p* < 0.001). Overall, there were no major differences in ASQ scores between the two groups.

Table 4 depicts the evaluation of child ASQ scores per domain at 3 years of age. The proportion of children from the IG and CG who had a Communication, Gross motor, or Personal-social score below the cut-off was comparable between the groups. Children from the IG were less likely to have a Fine motor score below the cut-off than children from the CG (1.2% vs. 2.5%, *p* = 0.002). However, the odds of children from the IG scoring below the cut-off for Problem-solving were 2.07 times that of children from the CG (*p* < 0.001).

## 4. Discussion

The purpose of this secondary analysis was to investigate the potential long-term effects of the GeliS antenatal lifestyle intervention, which is aimed at improving maternal health behaviour, on child anthropometrics and neurodevelopment up to 3 years of age. The results did not provide significant evidence for an intervention effect on the long-term anthropometrics and neurodevelopment of the offspring.

Child anthropometrics in the 2nd and 3rd years of life were comparable in the IG and CG. Though the proportion of children in the weight categories at 2 years of age were similar in the IG and CG, there was a trend for increased odds of being in a higher weight category for the children from the IG compared to the CG at 3 years of age. However, this might be due to a lower proportion of children from the IG in the underweight category.

The proportion of children with overweight was comparable to a German cohort that observed 6.2% of 3-to6-year-old children to be overweight [41]. Compared to this cohort data, however, we observed a higher rate of children with underweight, and a slightly lower rate of children with obesity (extremely underweight: 1.4%, underweight: 3.8%, obese: 2.9%) [41]. The missing evidence of a reduction in the rate of childhood overweight at 2 and 3 years of age by antenatal lifestyle counselling in the GeliS study is in accordance with findings from other studies [19,20,42,43,44], and supported by findings from two recent meta-analyses [23,24]. Louise et al. [24] performed an individual patient meta-analysis on randomised controlled trials (RCTs) including women with overweight or obesity on childhood outcomes at 3–5 years of age. Furthermore, a recent meta-analysis from our research group investigated RCTs including women from all BMI categories, and including offspring data from 1 month to 7 years of age [23]. Both meta-analyses sought to determine the effect of lifestyle interventions during pregnancy on offspring outcomes, and found no evidence of an alteration of child obesity risk and childhood weight or growth. In line with our findings, the LIMIT trial found no differences in child weight, height, and respective z-scores at 3–5 years of age following an antenatal lifestyle intervention in women with overweight and obesity [19]. These results are, furthermore, in agreement with results from the UPBEAT trial, which investigated group differences between comparable anthropometrics at 3 years of age in children born to mothers with obesity [43]. However, these comparable large-scale trials [19,43] have only considered women with overweight and/or obesity, whereas we also included women with normal weight. Furthermore, none of the aforementioned studies [19,20,42,43,44] were conducted alongside routine care.

One potential reason that we did not observe an intervention effect could be that both our GeliS [25] and the LIMIT [45] intervention were not successful in reducing GWG, a factor which has been linked to an increased risk of overweight/obesity in children [46]. Notably, the UPBEAT trial reported similar null findings on differences in child anthropometrics, despite the IG women having a slightly lower mean GWG [47]. Therefore, larger reductions in GWG, and the prevention of excessive GWG, might be necessary to improve offspring overweight/obesity risk, and this hypothesis should be further investigated.

Child neurodevelopment outcomes, assessed by the ASQ, were, overall, similar between the IG and CG. Children from the IG had a slightly higher risk for a potential delay in development in Problem-solving, and had a slightly lower risk for a potential delay in development in the Fine motor domain. Generally, the proportion of children scoring below the cut-off, and, therefore, being at risk for neurodevelopmental delay, was very low in our analysis. Due to small between-group differences in the above-mentioned categories, the clinical relevance is questionable, and should be interpreted with caution.

Results from the literature regarding antenatal lifestyle intervention effects on offspring neurodevelopment are mixed, and the comparability to our results is limited by heterogeneous study populations [19,20,21], varying measurement tools [20], and settings outside of routine care [19,20,21]. On one hand, our results are in line with findings from the LIMIT trial [19], which reported no evidence of an effect of a lifestyle intervention on ASQ domain scores at 3–5 years of age, and found scores comparable to our results in the individual domains [19]. The RCTs RADIEL and LIFEstyle, which both conducted maternal lifestyle interventions before and during pregnancy, also reported no differences in child neurodevelopmental scores between the IG and CG at their planned follow-up at 3–6 years of age [21]. On the other hand, Braeken and Bogaerts found significantly less surgency/extraversion in 3–7 year old offspring of the IG compared to CG following an antenatal lifestyle intervention conducted with brochures or brochures combined with prenatal lifestyle intervention sessions [20]. Interestingly, it has recently been hypothesised that interventions aimed at improving diet during pregnancy might only influence child executive function and behaviour in at-risk children from less favourable home environments [48]. Potentially, a similar relationship might partially explain the lack of effect of antenatal lifestyle interventions on child neurodevelopment observed by us and others [19,21]. However, we did not investigate the home environment, and are, therefore, unable to estimate if it modulated the intervention effect on neurodevelopment. Furthermore, though our GeliS intervention resulted in some minor improvements in the maternal diet in late pregnancy [26], and modestly affected breastfeeding behaviour [30], the impact was probably too small to affect child neurodevelopment at 3 years of age.

Our lack of evidence of an effect of our antenatal lifestyle intervention on child anthropometrics and neurodevelopment, coupled with inconsistent findings from other studies, suggests that the nature of the intervention is most decisive in improving long-term child health outcomes. However, the optimal intervention type and intensity are still unknown. A recent investigation indicated that passive interventions using brochures might be more successful in changing maternal lifestyle than brochures combined with active intervention sessions, possibly due to a higher intrinsic motivation [20]. This hypothesis is interesting, and indicates that less-intensive interventions could also have an effect. Furthermore, a recent meta-analysis has focused on the use of smartphone-based interventions to promote maternal health behaviours and maternal-foetal health outcomes, and indicated that multimodal interventions that also include another method of communication seem to have the highest effectiveness in reducing GWG [49]. Regarding smartphone applications (apps), non-time-consuming, concise, and practical information and advice are valued by participants [50], and apps may help overcome some barriers caused by socioeconomic gaps by being easily accessible to all women [50]. Overall, the above-mentioned aspects could be interesting approaches for the design of future trials, and might be able to improve child health outcomes by increasing the intervention effect on maternal health and lifestyle during pregnancy.

The current analysis has several limitations. We did not collect socioeconomic status in our trial, which may be associated with child development [51], and used maternal educational level as a proxy instead. In future trials, direct measures of socioeconomic status should be considered. Child anthropometric measurements were conducted alongside routine care, and thus, by varying personnel, which may limit the comparability due to a lack of standardisation. During the course of our trial, a new version of the well-baby check-up booklet was released that does not collect head circumference measurements at 3 years of age. This led to a lower number of available data on head circumference at that time-point. Despite officially defined time ranges for the two child health examinations in the 2nd and 3rd years of life, the target range was not always met. Therefore, we controlled for the varying timing of the child measurements by including the actual age at each examination as a confounding variable. Due to the utilization of a German reference group [37] and German recommendations [37], the comparability of our sex- and age-adjusted outcomes, as well as weight group assignment to international data, might be limited. The ASQ was filled out unsupervised by the mothers at home, and delays in answering the questionnaire could not be avoided. However, deviations from the proposed age at completion were accounted for by including the age at completion of the ASQ as a confounding factor in the analyses. Furthermore, it should be noted that the ASQ was not offered to the first 319 participants contacted in the 3-year follow-up.

Apart from these limitations, our study has several strengths worth noting. The GeliS study took place within the German routine care system as a cluster-randomised, controlled trial. Furthermore, it has a large sample size, originally comprising 2286 women from normal weight, overweight, and obese BMI categories, and a pre-planned 5-year follow-up phase. In this follow-up analysis 3 years after birth, we were able to include data from both the 2nd and 3rd years of life from 1644 mother-child pairs, which constitutes 71.9% of the original cohort. Compared to other lifestyle intervention trials [42,43], this represents a very high retention rate in the follow-up phase. Furthermore, the child anthropometric data originate from the children’s health records, and thus, represent data directly retrieved from the primary care setting. The well-baby check-up booklet provides clear guidelines for performing child health examinations. Using measurements collected during routine care enabled us to include data from a large number of children in our analysis. Although we observed no evidence of an effect of our intervention on child anthropometrics and neurodevelopment, the GeliS study will continue to follow-up the mother-child pairs until the 5th birthday, investigating potential delayed intervention effects, and collecting further valuable data on child development and lifestyle.

## 5. Conclusions

To our knowledge, the GeliS trial is the first large-scale trial with an antenatal lifestyle intervention conducted alongside routine care, and including women with normal weight, overweight, and obesity to report data on child anthropometrics and neurodevelopment in the 2nd and 3rd years of life. Despite small, but lasting, effects of the intervention on maternal lifestyle during pregnancy [26,27] and until the 1st year postpartum [28] and on the proportion of women breastfeeding exclusively [30], the current analysis detected no long-term effect of the intervention on child outcomes at 2 and 3 years of age. The continued GeliS follow-up will provide further valuable data on child growth development and health status until the age of 5 years, and offers the opportunity to identify factors influencing child health and overweight/obesity risk. Further studies using new innovative interventions may be helpful to effectively influence child health in the long-term.

## Figures and Tables

**Figure 1 jcm-11-01688-f001:**
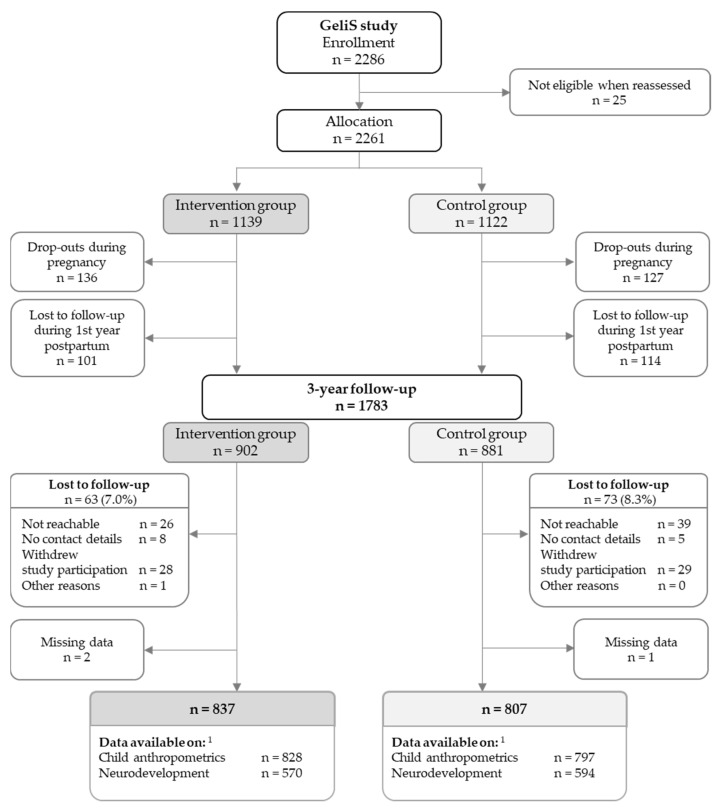
Flow of study participants. ^1^ Participants who provided data on both child anthropometrics and neurodevelopment: intervention group: n = 561; control group: n = 584.

**Table 1 jcm-11-01688-t001:** Characteristics of mother-child pairs.

	IG n = 837	CG n = 807	Total n = 1644
**Maternal characteristics**			
Pre-pregnancy age, years ^a^	30.5 ± 4.2	30.8 ± 4.3	30.6 ± 4.3
Pre-pregnancy weight, kg	68.4 ± 12.9	67.8 ± 13.3	68.1 ± 13.1
Pre-pregnancy BMI, kg/m^2^	24.4 ± 4.3	24.2 ± 4.5	24.3 ± 4.4
Pre-pregnancy BMI category, n (%)			
BMI 18.5–24.9 kg/m^2^	540/837 (64.5%)	537/807 (66.5%)	1077/1644 (65.5%)
BMI 25.0–29.9 kg/m^2^	201/837 (24.0%)	176/807 (21.8%)	377/1644 (22.9%)
BMI 30.0–40.0 kg/m^2^	96/837 (11.5%)	94/807 (11.6%)	190/1644 (11.6%)
GWG, kg	14.0 ± 5.2	14.0 ± 5.1	14.0 ± 5.1
GDM, n (%)	83/822 (10.1%)	68/761 (8.9%)	151/1583 (9.5%)
Educational level, n (%) ^b^			
General secondary school	103/836 (12.3%)	121/806 (15.0%)	224/1642 (13.6%)
Intermediate secondary school	364/836 (43.5%)	330/806 (40.9%)	694/1642 (42.3%)
High school	369/836 (44.1%)	355/806 (44.0%)	724/1642 (44.1%)
Country of birth, n (%)			
Germany	744/837 (88.9%)	732/806 (90.8%)	1476/1643 (89.8%)
Others	93/837 (11.1%)	74/806 (9.2%)	167/1643 (10.2%)
Primiparous, n (%)	532/837 (63.6%)	439/807 (54.4%)	971/1644 (59.1%)
Smoking in late pregnancy, n (%)	24/806 (3.0%)	30/780 (3.8%)	54/1586 (3.4%)
Current smoker, n (%) ^c^	103/743 (13.9%)	99/719 (13.8%)	202/1462 (13.8%)
**Infant characteristics at birth**			
Sex, n (%)			
Male	426/837 (50.9%)	439/807 (54.4%)	865/1644 (52.6%)
Female	411/837 (49.1%)	368/807 (45.6%)	779/1644 (47.4%)
Preterm birth, n (%)	52/834 (6.2%)	46/807 (5.7%)	98/1641 (6.0%)
SGA, n (%)	70/834 (8.4%)	61/807 (7.6%)	131/1641 (8.0%)
LGA, n (%)	64/834 (7.7%)	59/807 (7.3%)	123/1641 (7.5%)
Birth weight > 4000 g, n (%)	73/836 (8.7%)	66/807 (8.2%)	139/1643 (8.5%)

Abbreviations: IG: intervention group; CG: control group; BMI: body mass index; GDM: gestational diabetes mellitus; GWG: gestational weight gain; LGA: large for gestational age (>90th percentile); SGA: small for gestational age (<10th percentile); SD: standard deviation. ^a^ Mean ± SD (all such values). ^b^ General secondary school: General school, which is completed through year 9; Intermediate secondary school: Vocational secondary school, which is completed through year 10; High school: Academic high school, which is completed through year 12 or 13. ^c^ Collected 3 years after birth.

**Table 2 jcm-11-01688-t002:** Child anthropometrics at 2 and 3 years of age.

	Age	IG		CG		Adjusted Effect Size ^a^ (95% CI)	Adjusted *p* Value ^a^
		n	Mean ± SD	n	Mean ± SD		
Weight, kg	2 years	823	12.30 ± 1.44	790	12.26 ± 1.37	0.09 (−0.04, 0.23) ^b^	0.176
	3 years	818	14.58 ± 1.73	783	14.54 ± 1.72	0.12 (−0.05, 0.29)	0.166
Height, cm	2 years	821	87.0 ± 3.4	785	86.9 ± 3.3	0.27 (−0.05, 0.58)	0.098
	3 years	818	96.2 ± 3.9	779	96.1 ± 3.7	0.36 (−0.01, 0.73)	0.060
Head circumference, cm	2 years	814	48.4 ± 1.5	785	48.5 ± 1.4	0.01 (−0.13, 0.14)	0.933
	3 years	578	49.8 ± 1.5	515	49.8 ± 1.4	0.07 (−0.08, 0.21)	0.381
BMI, kg/m^2^	2 years	819	16.2 ± 1.4	783	16.2 ± 1.4	0.03 (−0.10, 0.17)	0.646
	3 years	817	15.7 ± 1.3	779	15.7 ± 1.3	0.03 (−0.10, 0.16)	0.663
Weight z-score ^c^	2 years	823	0.07 ± 0.88	790	0.02 ± 0.84	0.05 (−0.04, 0.13)	0.254
	3 years	818	0.14 ± 0.88	783	0.11 ± 0.88	0.04 (−0.04, 0.13)	0.312
Weight percentile ^c^	2 years	823	51.8 ± 26.8	790	50.6 ± 26.2	1.48 (−1.12, 4.08)	0.263
	3 years	818	54.3 ± 26.7	783	53.1 ± 26.3	1.41 (−1.20, 4.01)	0.290
Height z-score	2 years	821	0.06 ± 0.96	785	0.00 ± 0.91	0.07 (−0.02, 0.16)	0.153
	3 years	818	0.16 ± 0.98	779	0.10 ± 0.93	0.06 (−0.03, 0.16)	0.188
Height percentile	2 years	821	51.7 ± 28.1	785	49.9 ± 27.3	2.03 (−0.71, 4.76)	0.146
	3 years	818	54.4 ± 27.8	779	52.5 ± 27.5	2.15 (−0.57, 4.88)	0.121
BMI z-score	2 years	819	0.03 ± 1.00	783	0.03 ± 0.98	0.01 (−0.09, 0.11)	0.808
	3 years	817	0.00 ± 0.93	779	−0.01 ± 0.95	0.01 (−0.08, 0.11)	0.778
BMI percentile	2 years	819	51.1 ± 28.6	783	51.2 ± 28.4	0.11 (−2.68, 2.91)	0.936
	3 years	817	50.2 ± 27.4	779	50.2 ± 28.0	0.00 (−2.72, 2.72)	1.000
**Weight category**	2 years	**n (%)**		**n (%)**		**Adjusted Effect** **Size ^d^ (95% CI)**	**Adjusted** ***p* value ^d^**
Underweight (<10th BMI percentile)		73/819 (8.9%)		79/783 (10.1%)		1.15 (0.89, 1.49)	0.274
Normal weight (10th–90th BMI percentile)		654/819 (79.9%)		627/783 (80.1%)			
Overweight (>90th–97th BMI percentile)		70/819 (8.5%)		57/783 (7.3%)			
Obesity (>97th BMI percentile)		22/819 (2.7%)		20/783 (2.6%)			
**Weight category**	3 years	**n (%)**		**n (%)**			
Underweight (<10th BMI percentile)		62/817 (7.6%)		74/779 (9.5%)		1.17 (0.99, 1.37)	0.062
Normal weight (10th–90th BMI percentile)		683/817 (83.6%)		644/779 (82.7%)			
Overweight (>90th–97th BMI percentile)		57/817 (7.0%)		48/779 (6.2%)			
Obesity (>97th BMI percentile)		15/817 (1.8%)		13/779 (1.7%)			

Abbreviations: IG: intervention group; CG: control group; BMI: body mass index; CI: confidence interval; SD: standard deviation. ^a^ From mixed models for repeated measures with the use of data from each health examination from the 1st to the 3rd year of life, and controlled for maternal pre-pregnancy age, maternal pre-pregnancy BMI category, parity, child sex, child age in days at the corresponding visit, and study region. ^b^ Estimated mean difference; in parentheses, 95% CI (all such values). ^c^ All z-scores and percentiles were calculated according to Kromeyer-Hauschild et al. [37]. ^d^ From proportional odds ordinal logistic regression models fit using generalised estimating equations controlled for maternal pre-pregnancy age, maternal pre-pregnancy BMI category, and parity.

**Table 3 jcm-11-01688-t003:** Child neurodevelopment assessed by ASQ scores in the five domains at 3 years of age.

	IG		CG		Adjusted Effect Size ^a^ (95% CI)	Adjusted *p* Value ^a^
	n	Mean ± SD	n	Mean ± SD		
Communication	567	55.6 ± 6.2	589	55.1 ± 6.7	0.28 (−0.50, 1.06)	0.488
Gross motor	567	55.3 ± 7.0	594	54.8 ± 8.0	0.57 (−0.34, 1.49)	0.217
Fine motor	563	50.2 ± 10.7	592	50.0 ± 11.2	−0.39 (−1.18, 0.40)	0.335
Problem-solving	561	54.3 ± 8.1	589	54.9 ± 7.2	−0.67 (−0.97, −0.37)	<0.001
Personal-social	569	53.7 ± 6.3	594	53.9 ± 6.4	−0.11 (−0.79, 0.57)	0.744

Abbreviations: ASQ: Ages and Stages Questionnaire (ASQ-3^TM^); IG: intervention group; CG: control group; CI: confidence interval; SD: standard deviation; BMI: body mass index. ^a^ Linear regression models fit using generalised estimating equations adjusted for maternal pre-pregnancy age, maternal pre-pregnancy BMI category, maternal educational level, parity, child sex, child age in months at completion of the ASQ.

**Table 4 jcm-11-01688-t004:** Proportion of children with ASQ scores below cut-off at 3 years of age.

	IG		CG		Adjusted Odds Ratio ^a^ (95% CI)	Adjusted *p* Value ^a^
	n	%	n	%		
Communication	6/567	1.1%	9/589	1.5%	0.73 (0.32, 1.70)	0.470
Gross motor	19/567	3.4%	26/594	4.4%	0.73 (0.38, 1.40)	0.342
Fine motor	7/563	1.2%	15/592	2.5%	0.45 (0.28, 0.74)	0.002
Problem-solving	14/561	2.5%	10/589	1.7%	2.07 (1.45, 2.95)	<0.001
Personal-social	9/569	1.6%	10/594	1.7%	1.31 (0.78, 2.19)	0.303

Depicted is the proportion of children from the IG and CG with ASQ scores below cut-off in the individual domains. Abbreviations: ASQ: Ages and Stages Questionnaire (ASQ-3^TM^); IG: intervention group; CG: control group; CI: confidence interval; BMI: body mass index. ^a^ Binary logistic regression models fit using generalised estimating equations adjusted for maternal pre-pregnancy age, maternal pre-pregnancy BMI category, maternal educational level, parity, child sex, child age in months at completion of the ASQ.

## Data Availability

The datasets used and analysed during the current study are available from the corresponding author on reasonable request.

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
