# Peer review of "Child Anthropometrics and Neurodevelopment at 2 and 3 Years of Age Following an Antenatal Lifestyle Intervention in Routine Care—A Secondary Analysis from the Cluster-Randomised GeliS Trial"

_jcm, 2022, doi:10.3390/jcm11061688_

Round 1

Reviewer 1 Report

Thank you to the authors for their time in considering my comments and making changes. Unfortunately, I cannot easily write responses to your comments in-line given the limitations of this system, and hope the following might be easy to understand. 

Background: No further comment. 

Methods: (1)Thank you for the context. My stance was from the point of someone doing a future systematic review and meta-analysis, wherein not including the WHO growth values could limit the inclusion of your study. No further change necessary. (2) My point was actually around how you had taken into consideration gestational age at birth in correcting postnatal age, and whether this affected your findings. The comment was not specific to preterm infants alone, or their exclusion, or adjusting for preterm births. Seeing as you have now run an analysis investigating the effect of preterm birth on the outcomes of interest, you could state something along the lines of "In a sensitivity analysis, the exclusion of preterm births did not affect estimates (data not shown)."  (3) 

Results: (5) I think it is worth noting in the Discussion with the limitations that beyond maternal education SES data was not collected. Yes, maternal education can serve as a proxy for SES, but a proxy is not a direct measure, and a future direction could be collecting further, more direct SES measures.  (6) That is good to see that compliance was high with counselling. This rate (98.2%) should be mentioned in the manuscript, as other readers will likely have the same question that I did about exposure. 

Discussion: No further comment (beyond items in #5 above).

Reviewer 2 Report

Thank you very much to the authors of the manuscript for referring and taking into account my suggestions. Your manuscript has certainly become clearer to the reader. 

Author Response

This manuscript is a resubmission of an earlier submission. The following is a list of the peer review reports and author responses from that submission.

Round 1

Reviewer 1 Report

Thank you for the opportunity to review this well-written manuscript describing anthropometric and neurodevelopmental outcomes among children born to mothers in the GeliS Trial. While no improvement in outcomes of interest was observed at 2- and 3-years following an antenatal care-based lifestyle intervention, this is still of interest to the broader research community. As described by the authors, such findings appear consistent with the literature to date. Below, please find some suggestions that could improve the manuscript. 

Background: 
- As written, the background is appropriate and well stated. Within paragraphs 3 and 4, two sentences each start with "so far" and "however", thus the authors could vary consider varying the sentence structure. 

Methods: 
- Have the authors considered presenting the anthropometric cut-offs using the WHO growth standards? The rationale for using the German standards is presented, and makes sense, but to extend the interpretability of the anthropometrics to an international audience, including the WHO-GS cut-offs would be helpful. This could be placed in an appendix. 
- Perhaps I missed it, but were there any steps taken to correct for gestational age at birth? This could affect the interpretability of growth measures, particularly if an infant were preterm. The authors might consider the following paper: 
Nandita Perumal, Daniel E Roth, Donald C Cole, Stanley H Zlotkin, Johnna Perdrizet, Aluisio J D Barros, Ina S Santos, Alicia Matijasevich, Diego G Bassani, Effect of Correcting the Postnatal Age of Preterm-Born Children on Measures of Associations Between Infant Length-for-Age z Scores and Mid-Childhood Outcomes, American Journal of Epidemiology, Volume 190, Issue 3, March 2021, Pages 477–486, https://doi.org/10.1093/aje/kwaa169

Results: 
- Figure 1: As presented, it is unclear how many children had both anthro and neurodevelopmental data at the 3-year follow up point (e.g., in the IG, 837 children did not have both data available). Perhaps this could be clarified using a subscript? Or a further breakdown could be provided (i.e., both available, only anthro available, only neurodevelopment available)?
- Were there any data collected on socioeconomic status of the mothers? Was this considered in the model generation process? 
- Did all mothers in the intervention group receive all exposures (i.e., counselling sessions) to the intervention programme? Is there any way to measure fidelity of the intervention provided to those in the intervention group? Perhaps this might be considered as a covariate in the model, or considered as a sensitivity analysis. 

Discussion: 
- Overall, the Discussion is clear. Some of the paragraphs are a bit long, and the first paragraph in particular would benefit from being broken up into shorter paragraphs. 
- It is interesting that the difference in BMI categories between groups is particularly among those underweight. That is, within the control group, there are more underweight children. Although the total magnitude of the difference is small, do the authors have any suggestions as to why this is? This also relates to why I was curious about socioeconomic status. It does appear more mothers in the control group had secondary education only. 

Reviewer 2 Report

Review of the manuscript entitled ‘Child anthropometrics and neurodevelopment at 2 and 3 years of age following an antenatal lifestyle intervention in routine care – a secondary analysis from the cluster-randomised GeliS trial’ (Manuscript ID: jcm-1495018). The work is properly prepared, however, from the point of view of a reviewer, I would like to request for a few minor corrections:

  1. In general, the introduction to the study is correct. However, the authors are asked to place particular emphasis to the knowledge gaps that the study fills.
  2. In spite of the fact that the protocol was described in other part of the study, the authors are required to clarify the meaning of IG and CG in the methodology. The aforementioned abbreviations appear in the methodology, however there is no explanation that the studied women were divided into two groups. In the further part of the paper, the authors are asked to follow the adopted abbreviations, i.e., if the authors decide to introduce abbreviations, there is no need to use their full names, like in Table No 1.
  3. With regard to the results presented in Table No 2, I believe that the analyses based on the averages obtained should be included in the main body of the text, while the analyses based on the categories should be transferred to the supplementary materials, obviously referring to them in the main part of the study and indicating that the reader will find such information in the supplementary materials.
  4. The reference list of sources should meet editorial requirements.